# Chemical Composition, Bioactivities, and Applications of Spirulina (*Limnospira platensis*) in Food, Feed, and Medicine

**DOI:** 10.3390/foods13223656

**Published:** 2024-11-17

**Authors:** Maria P. Spínola, Ana R. Mendes, José A. M. Prates

**Affiliations:** 1CIISA—Centro de Investigação Interdisciplinar em Sanidade Animal, Faculdade de Medicina Veterinária, Universidade de Lisboa, Av. da Universidade Técnica, 1300-477 Lisboa, Portugal; mariaspinola@fmv.ulisboa.pt (M.P.S.); rmendes@isa.ulisboa.pt (A.R.M.); 2Associate Laboratory for Animal and Veterinary Sciences (AL4AnimalS), Av. da Universidade Técnica, 1300-477 Lisboa, Portugal; 3LEAF—Linking Landscape, Environment, Agriculture and Food Research Centre, Instituto Superior de Agronomia, Universidade de Lisboa, Tapada da Ajuda, 1349-017 Lisboa, Portugal

**Keywords:** *Limnospira platensis*, bioactive compounds, antioxidant activity, anti-inflammatory activity, immunomodulatory effects, food, feed, medicine

## Abstract

Spirulina (*Limnospira platensis*) is a microalga recognised for its rich nutritional composition and diverse bioactive compounds, making it a valuable functional food, feed, and therapeutic agent. This review examines spirulina’s chemical composition, including its high levels of protein, essential fatty acids, vitamins, minerals, and bioactive compounds, such as the phycocyanin pigment, polysaccharides, and carotenoids, in food, feed, and medicine. These compounds exhibit various biological activities, including antioxidant, anti-inflammatory, immunomodulatory, antiviral, anticancer, antidiabetic and lipid-lowering effects. Spirulina’s potential to mitigate oxidative stress, enhance immune function, and inhibit tumour growth positions it as a promising candidate for preventing chronic diseases. Additionally, spirulina is gaining interest in the animal feed sector as a promotor of growth performance, improving immune responses and increasing resistance to diseases in livestock, poultry, and aquaculture. Despite its well-documented health benefits, future research is needed to optimize production/cultivation methods, improve its bioavailability, and validate its efficacy (dose–effect relationship) and safety through clinical trials and large-scale human trials. This review underscores the potential of spirulina to address global health and nutrition challenges, supporting its continued application in food, feed, and medicine.

## 1. Introduction

The blue-green spirulina (*Limnospira platensis*) microalga, formerly *Arthrospira platensis* [1], has gained significant attention in recent decades due to its diverse bioactive compounds and potential health benefits [2]. Spirulina is rich in essential nutrients, including proteins, vitamins, minerals, and antioxidants, which have been attributed to a wide range of therapeutic effects, such as immune modulation, and anti-inflammatory and anticancer properties [3,4,5,6]. It is particularly known for its high protein content and can compete with conventional protein sources such as soybeans or eggs, making it an excellent candidate for addressing protein deficiencies in developing regions [7]. Additionally, the presence of unique pigments such as phycocyanin has generated interest in its applications in the pharmaceutical, nutraceutical, and food industries [8]. In the genus *Arthrospira*, *A. platensis* and *A. maxima* are the most studied and recognized species, with scientific evidence of their bioactive compounds and effects, and they are generally considered safe for human consumption [9]. *A. fusiformis* has already been studied to optimize the synthesis of photosynthetic pigments, but other nutrients have not been considered [10]. For the species *A. laxissima,* there is still a lack of information concerning its bioactive compounds in higher amounts, beneficial effects, and beneficial properties for human health [11].

Research has also explored spirulina’s role as a powerful antioxidant [12]. Several studies suggest that the phenolic compounds, flavonoids, and phycocyanins present in spirulina can mitigate oxidative stress and prevent chronic diseases such as cancer and diabetes [13]. These bioactive compounds are not only nutritionally valuable but have also shown promising results in the treatment of viral infections and possess anticancer and anti-inflammatory properties [14]. However, the effects of environmental and processing conditions on the potency of these bioactive compounds remain a subject of debate. For instance, thermal treatment can degrade phycocyanin, affecting its functional properties [15]. Similarly, the role of fermentation in enhancing spirulina’s bioactive profile has raised questions regarding the optimal conditions for maximizing its nutraceutical potential [16].

Given the increasing demand for sustainable and functional food sources, spirulina presents itself as a promising solution for nutritional supplementation and pharmaceutical applications [16]. This systematic review comprehensively assesses the chemical composition and bioactive compounds of spirulina and evaluates its potential applications in food, animal feed, nutrition, and medicine. To ensure a thorough and precise review, we conducted a systematic search using key databases, including Google Scholar (Google LLC, Mountain View, CA, USA), PubMed (NCBI, Bethesda, MD, USA), Scopus (Elsevier B.V., Amsterdam, The Netherlands), and Web of Science (Clarivate Analytics, Philadelphia, PA, USA).

Our search strategy involved the use of specific search terms (e.g., “Spirulina”, “bioactive compounds”, “phycocyanin”, and “antioxidant activity”) and applied inclusion and exclusion criteria, selecting peer-reviewed studies and focusing on recent research. The bibliometric analysis provided valuable insights into the research landscape. The search results varied significantly across databases: Google Scholar yielded the highest number of review articles, with nearly 30,000, followed by Web of Science with 950, Scopus with 708, and PubMed with 335 articles. The keyword “Spirulina” generated more results compared to others, while the term “antioxidant activity” returned 130 review articles on the Web of Science. In PubMed, articles referencing “Spirulina” date back to 1976; however, recent research has increasingly focused on bioactive compounds, particularly phycocyanin. Over the last decade, there have been 42 review articles referring to “phycocyanin” in PubMed, with 34 of them published in the past five years (2021–2024), highlighting the growing interest in its potential health benefits. To minimize redundancy, only the most recent articles were selected for duplicated entries across different databases.

This review synthesizes current evidence on spirulina’s bioactive compounds, including their antioxidant, anti-inflammatory, immunomodulatory, antiviral, anticancer, antidiabetic, and lipid-lowering properties. Additionally, the study explores functional and therapeutic applications, identifies existing knowledge gaps, and outlines future research and industrial utilization opportunities.

## 2. Chemical Composition of Spirulina

Spirulina is a microalga well known for its complex and nutritionally rich chemical composition, which makes it a valuable organism for various food, pharmaceutical, nutraceutical, and animal feed applications [17]. The composition of spirulina is primarily characterized by its high content of proteins, lipids, carbohydrates, vitamins, minerals, and pigments, all of which contribute to its biological activities [18].

### 2.1. Proteins

The most abundant component of spirulina is protein, which accounts for 50–70% of its dry weight [19,20]. This high protein content is particularly notable because spirulina contains all essential amino acids, making it a complete protein source comparable to that found in animal products such as meat and eggs [18]. Essential amino acids (38.81%) include leucine (7.67%), lysine (4.37%), methionine (2.39%), phenylalanine (4.42%), threonine (4.88%), tryptophan (1.93%), and valine (6.37%) [21,22]. Non-essential amino acids (61.19%) such as alanine (7.52%), arginine (7.65%), glycine (5.24%), and serine (4.16%) are also present in significant amounts, further increasing the quality of the protein profile of spirulina [22,23]. This makes spirulina an attractive source of dietary protein, especially for vegetarian and vegan populations [21].

One of the primary challenges of using spirulina as a protein ingredient, particularly at higher incorporation levels in diets, is its rigid peptidoglycan cell wall [24], which makes the digestibility, bioaccessibility, and bioavailability of its nutrients difficult [24,25,26]. The proteins in spirulina are often bound in protein–pigment complexes within the thylakoid membrane, making them difficult to access without adequate pre-treatment [15,27]. Studies have demonstrated that mechanical and physical pre-treatments, such as bead milling, extrusion, freeze-drying, heating, microwave, and sonication, can significantly enhance the solubility and degradation of these proteins [25,26]. All of these pre-treatments have advantages and disadvantages, and they are described in Table 1. Of these methods, extrusion has proven to be especially effective at disrupting the cell wall and increasing protein bioaccessibility, with or without the combination of enzymes [25,26]. By denaturing and aggregating key proteins like phycocyanin, extrusion facilitates greater nutrient availability for digestion [17]. This approach may allow spirulina to be incorporated at higher levels in diets [28,29,30].

### 2.2. Lipids

Lipids in spirulina are present in lower quantities than proteins but still play an important role in its overall chemical composition [23]. The lipid fraction represents approximately 5–10% of the dry biomass, consisting primarily of polyunsaturated fatty acids (PUFAs) [19,40]. The most prominent fatty acid in spirulina is γ-linolenic acid (GLA; 18:3n − 6), an omega-6 fatty acid known for its anti-inflammatory properties [41]. Other important fatty acids include linoleic acid (18:2n-6), oleic acid (18:1c9), and palmitic acid (16:0) [17,18]. The GLA content in spirulina can vary depending on cultivation conditions but typically ranges between 1 and 2% of the total dry weight. Additionally, spirulina contains smaller amounts of omega-3 fatty acids such as α-linolenic acid (ALA; 18:3n-3), although these are present in much lower concentrations compared to GLA and other PUFAs [18].

### 2.3. Carbohydrates

Carbohydrates constitute approximately 15–20% of the dry weight of spirulina [42]. These carbohydrates are primarily in the form of polysaccharides, which serve both as structural components and as storage molecules within the cyanobacterium [43]. Notably, spirulina contains sulphated polysaccharides that are known for their bioactive properties, including antiviral, anti-inflammatory, and immune-modulating activities [44]. The main monosaccharides found in spirulina include glucose, rhamnose, xylose, mannose, and galactose [44]. These polysaccharides form complex structures, which contribute to the cell wall integrity and potentially impact the organism’s bioactivities [45]. Sulphated polysaccharides, including calcium spirulan (Ca-SP), a very unique sulphated polysaccharide, are also present in its cell wall [46]. In addition, spirulina contains lower-molecular-weight carbohydrates, including glycogen, which serve as an energy reserve [47].

### 2.4. Pigments

One of the defining characteristics of spirulina is its rich pigment content, which is responsible for its distinctive blue-green colour [48]. The primary pigment in spirulina is phycocyanin, a blue pigment–protein complex that can account for up to 47% of its dry weight [23,49]. Phycocyanin plays a key role in the light-harvesting complexes of photosynthesis and is also recognized for its potent antioxidant properties [42]. Beyond phycocyanin, spirulina contains other pigments, such as chlorophyll and carotenoids, vitamin B and vitamin E [50]. Chlorophyll is essential for photosynthesis and is present in significant quantities, contributing to spirulina’s ability to thrive in various aquatic environments [48,51]. The carotenoids found in spirulina not only assist in photosynthetic activity but also act as antioxidants, neutralizing free radicals and contributing to the organism’s overall health-promoting potential [49].

### 2.5. Vitamins and Minerals

Spirulina is a rich source of essential vitamins and minerals [18,50]. It contains high levels of B-complex vitamins, particularly thiamine (B1), riboflavin (B2), niacin (B3), and folate (B9), all of which are critical for energy metabolism and cellular function [23,52]. Vitamin B12 is also present, though its bioavailability and activity are subject to ongoing research and debate [52,53]. In addition, spirulina is an excellent source of fat-soluble vitamins, particularly vitamin E (α-tocopherol) and vitamin A in the form of β-carotene, which the body can convert into active retinol [23,52].

Minerals are abundant in spirulina, with iron being one of the most significant [18]. The iron content in spirulina can range between 28 and 50 mg per 100 g of its dry weight, making it a valuable source of this mineral, especially for individuals with iron deficiency anaemia [53,54]. The other minerals present include calcium, magnesium, zinc, and potassium, all of which play vital roles in maintaining cellular homeostasis and supporting metabolic processes [53]. The bioavailability of these minerals in spirulina is generally high, enhancing its nutritional value as a supplement or food ingredient.

### 2.6. Secondary Metabolites

In addition to its major components, spirulina contains a variety of secondary metabolites that contribute to its bioactivity [23]. These include polyphenols, sterols, and phenolic acids, which exert significant biological effects [55]. Phenolic acids make up one-third of the phenolic compounds while flavonoids make up the remaining amount [21].

Some of the phenolic chemicals that might be found in spirulina are pyrogallol, gallic, chlorogenic caffeine, vanillic, p-coumaric, naringin, hespirdin, rutin, quercetrin, naringenin, catechin, and hespirtin [21,56,57].

For example, polyphenolic compounds, such as ferulic acid and caffeic acid, have been identified in spirulina and are known for their antioxidant, anti-inflammatory, and antimicrobial properties [55,58,59,60].

Phenolic compounds are molecules containing a benzene ring with at least one hydroxyl substituent and are the main example of bioactive compounds found in products of plant origin, including algae [23,61].

The presence of phenolic compounds augments the pharmaceutical properties of spirulina, such as its anticarcinogenic, antimicrobial, anti-inflammatory, and antitumoral effects [21,62,63]. A study developed by Abdel-Moneim et al. [64] analysed the antioxidant and antimicrobial activities of spirulina extracts against selected pathogenic bacteria and fungi. The effectiveness of antimicrobial activity was attributed to the methanol extract, which had a higher total phenolic content. 

## 3. Bioactivity of Spirulina Compounds

Spirulina has received attention for its diverse range of bioactive compounds, which exhibit numerous health-promoting properties [23,65]. These bioactivities are primarily attributed to its proteins, polysaccharides, fatty acids, pigments, and variety of secondary metabolites [23,48,66]. The therapeutic potential of these compounds includes antioxidant, anti-inflammatory, immunomodulatory, antiviral, anticancer, antidiabetic, and lipid-lowering effects [23,65,66,67,68,69,70]. 

### 3.1. Antioxidant Activity

The antioxidant potential of spirulina is one of its most widely studied bioactivities [67]. Its high levels of pigments, particularly phycocyanin, carotenoids, and chlorophyll, are key contributors to this activity [23,48]. Phycocyanin, the blue pigment in spirulina, has demonstrated potent scavenging effects against free radicals such as superoxide anions, hydroxyl radicals, and peroxyl radicals [65]. This antioxidant capacity helps mitigate oxidative stress, which is implicated in the pathogenesis of chronic diseases like cancer, cardiovascular diseases, and neurodegenerative disorders [71].

Carotenoids such as β-carotene, zeaxanthin, and lutein, present in spirulina, also contribute to its antioxidant properties [51]. These compounds neutralize singlet oxygen and other reactive oxygen species (ROS), protecting cells from oxidative damage [66]. Additionally, chlorophyll, a green pigment, has been shown to possess antioxidant activities that support detoxification processes and protect against oxidative injury in tissues [72].

A study developed by Abdel-Moneim et al. [64] analysed the antioxidant and antimicrobial activities of spirulina extracts against selected pathogenic bacteria and fungi. The extract with the most effective antimicrobial activity was the methanol extract, which had a high total phenolic content. All of the spirulina extracts had an antioxidant effect, which was the strongest in the methanol extract because of its high total phenolic content.

### 3.2. Anti-Inflammatory Effects

Spirulina contains bioactive compounds with significant anti-inflammatory properties, which can help in the management of inflammatory conditions [67,68]. Phycocyanin, in particular, has been shown to inhibit the production of pro-inflammatory cytokines like tumour necrosis factor-alpha (TNF-α) and interleukin-6 (IL-6), as well as reduce the activity of inflammatory enzymes such as cyclooxygenase-2 (COX-2) [73]. 

A study by Jiang et al. [74] showed that spirulina enriched with selenium may decrease cytokine levels, such as IL-6, IL-1β, and TNF-α. These combined anti-inflammatory actions make spirulina a promising candidate for preventing and managing inflammatory diseases. 

### 3.3. Immunomodulatory Activity

The immunomodulatory properties of spirulina have been widely studied, with evidence suggesting that it can enhance both innate and adaptive immune responses [23,67]. Phycocyanin plays a central role in modulating immune functions by enhancing the activity of natural killer (NK) cells, macrophages, and T cells, while also promoting the production of antibodies [44]. 

Spirulina’s polysaccharides also play a role in enhancing immune responses [23,75]. They stimulate the production of cytokines such as interferon-gamma (IFN-γ) and IL-2 (IL-2), which are crucial for activating immune cells and coordinating the immune response [76]. This immune stimulatory effect has led to research exploring spirulina’s potential to support immune function in patients with infections, cancer, and autoimmune diseases [75].

### 3.4. Antiviral Activity

Spirulina has demonstrated antiviral properties, particularly against enveloped viruses such as the herpes simplex virus, human cytomegalovirus, influenza A, human immunodeficiency virus (HIV), and hepatitis C [69]. A sulphated polysaccharide isolated from spirulina, Ca-SP, has been shown to inhibit viral replication by preventing the penetration of viruses into host cells [46]. The ability of Ca-SP to block viral entry makes it a promising candidate for antiviral therapies, especially in the context of emerging viral infections [46].

Phycocyanin has also been reported to exhibit antiviral activity, possibly by reducing viral replication rates, affecting RNA synthesis in vitro [69]. Also, spirulina extract has been described as a potential therapeutic agent, due to its action in the early stages of infection with the influenza virus, by reducing virus yields and inhibiting influenza hemagglutination [77]. These findings suggest that spirulina could be an adjunct in antiviral therapies, although further clinical studies are necessary to confirm these effects [77].

### 3.5. Anticancer Activity

Various research groups have developed work on spirulina’s anticancer compound properties, and the evidence concerning this subject is increasing [66,78]. Phycocyanin has been identified as a key player in spirulina’s anticancer effects [65]. This bioactive compound induces apoptosis (programmed cell death) in cancer cells, inhibits tumour cell proliferation, and suppresses angiogenesis (the formation of new blood vessels that supply tumours) [65,79]. Phycocyanin exerts these effects by activating pro-apoptotic pathways, such as the activation of caspases, and by inhibiting survival pathways, such as those mediated by nuclear factor kappa B (NF-κB) [65,80].

Moreover, carotenoids and polysaccharides from spirulina also contribute to its anticancer properties [51]. Carotenoids such as β-carotene have been shown to prevent DNA damage and reduce the risk of carcinogenesis [81]. Spirulina polysaccharides have exhibited anti-proliferative effects on gastric cancer cells by inducing apoptosis and displaying antimetastatic activity [82]. These compounds make spirulina a promising candidate for cancer prevention and treatment [82].

### 3.6. Antidiabetic Effects

Spirulina has been recognized for its potential antidiabetic effects, particularly in improving glycaemic control and modulating lipid profiles [66]. Key bioactive compounds such as phycocyanin, polysaccharides, and GLA have been shown to regulate blood glucose levels and enhance insulin resistance [65]. Studies indicate that spirulina supplementation can reduce fasting blood glucose and HbA1c levels, thus contributing to better management of type 2 diabetes [83]. These effects are largely attributed to its ability to modulate glucose metabolism by reducing its levels in the blood and enhancing insulin resistance [65].

### 3.7. Lipid-Lowering and Cardiovascular Effects

Spirulina is also recognized for its lipid-lowering properties [83]. Spirulina’s consumption has been associated with reductions in total cholesterol, low-density lipoprotein (LDL) cholesterol, and triglyceride levels, while increasing high-density lipoprotein (HDL) cholesterol [70]. These effects are largely attributed to phycocyanin and GLA, which help regulate lipid metabolism and prevent the accumulation of fats in blood vessels [70]. Additionally, spirulina’s antioxidant and anti-inflammatory properties contribute to the prevention of atherosclerosis and other cardiovascular diseases [84].

Table 2 summarizes spirulina’s key chemical compounds and their bioactivities, highlighting the diverse health benefits of this microalga.

## 4. Applications of Spirulina in Food, Feed, and Medicine

Spirulina microalga has been widely recognized for its broad range of applications in feed, food, and medicine, due to its rich nutritional profile (its high protein content, as well as essential fatty acids, vitamins, and minerals) and bioactive compounds [87]. 

### 4.1. Applications in Food

Spirulina is extensively used in the food industry as a dietary supplement and functional food ingredient due to its high nutritional value [48,50]. Spirulina is commonly found in powdered or tablet forms and added to smoothies, energy bars, and other food products [16,48]. Additionally, it is used as a natural colourant due to its high phycocyanin content, which provides a vivid blue pigment used in confections, beverages, and health products [87].

The addition of spirulina to food products is also driven by its beneficial health effects. Its rich supply of essential amino acids, vitamins (especially B-complex vitamins and β-carotene), and minerals like iron and calcium contribute to its enhanced nutritional value [18,23]. The application of spirulina in foods can improve nutritional intake, particularly in regions where malnutrition and nutrient deficiencies are prevalent [21]. Moreover, spirulina-enriched foods have been shown to lower blood cholesterol and blood sugar levels, contributing to better management of cardiovascular and metabolic health [21].

The antioxidant properties of spirulina also make it a valuable ingredient in functional foods aimed at reducing oxidative stress [65,67]. By incorporating spirulina into everyday food products, it offers protection against chronic diseases, including heart disease, cancer, skin disease, and arthritis [48,49]. Furthermore, spirulina’s antimicrobial properties have led to its inclusion in food preservation, enhancing the shelf-life of perishable products [86].

### 4.2. Applications in Feed

Spirulina has gained significant attention as a valuable ingredient in animal feed due to its rich nutritional composition and bioactive properties [17,19,41]. Spirulina is used as a dietary supplement for livestock, poultry, aquaculture, and even pet animals, offering benefits such as enhanced growth performance, improved immune function, and increased resistance to diseases [19,88].

Spirulina has been incorporated into livestock and poultry feed as a protein-rich supplement [6,29,89]. This microalga contains all essential amino acids required for animal growth and development, making it an ideal substitute for traditional protein sources like soybean meal and fish meal [7,41]. Studies have shown that adding spirulina to the diets of chickens, pigs, goats, and cattle improves their weight gain, feed conversion efficiency, and overall growth performance [28,41,89,90]. According to Costa et al. [28], extruded spirulina showed an increase in body weight and feed intake, compared to spirulina without pre-treatment (1183 g to 1349 g for body weight and 219 g to 254 g for feed intake). In Al-Yahyaey et al. [90]’s study, goats taking spirulina (four g/head/day) had an improved feed conversion ratio (15.15 in the control group and 9.79 in the group with spirulina). Additionally, spirulina enhances the nutritional quality of animal products, increasing the content of beneficial fatty acids in meat, eggs, and milk [91,92].

Spirulina supplementation has also been shown to enhance the immune response in livestock and poultry [91,93]. For instance, chickens fed with spirulina-enriched diets demonstrated improved resistance to bacterial and viral infections, as well as better vaccination responses [91]. This immunomodulatory effect is attributed to spirulina’s bioactive compounds, including phycocyanin, which helps modulate immune cell activity [44].

In aquaculture, spirulina is used as a high-quality feed additive for fish and shrimp due to its high protein content, essential fatty acids, vitamins, and minerals [94]. Fish species such as tilapia, carp, and salmon have shown improved growth rates, better feed utilization, and enhanced pigmentation when supplemented with spirulina [95,96,97,98]. When 7.5–10% spirulina was added to the diets of *Mustus cavasuis*, their body weight, specific growth rate, and feed conversion ratio were improved compared to a control group [98]. The natural pigments found in spirulina, including carotenoids like astaxanthin, help enhance the colouration of fish and improve the aesthetic quality of fish skin and flesh, which is particularly important in ornamental fish and commercial aquaculture [88]. Furthermore, spirulina improves the immune function of aquatic species, increasing their resistance to diseases and reducing the need for antibiotics and other chemical treatments in aquaculture systems [98]. 

Spirulina is increasingly used as a supplement in pet food due to its ability to support overall health and well-being in companion animals [99]. Dogs can benefit from spirulina’s rich nutrient profile, which supports immune health, digestion, and coat quality [99]. Its antioxidant properties help reduce inflammation and oxidative stress, contributing to better health outcomes in ageing pets and those with chronic conditions [68,71]. Additionally, spirulina has been shown to enhance the palatability of pet food, making it an attractive ingredient for pet owners looking to improve their pets’ nutrition [100].

### 4.3. Applications in Medicine

Spirulina has shown potential in therapeutic applications, ranging from immunomodulation to cancer prevention [23,65,66,67,68,69,70]. Its bioactive compounds, such as phycocyanin, sulphated polysaccharides, and PUFAs, contribute to its various medicinal benefits [23,42,78]. These compounds have been extensively studied for their anti-inflammatory, antioxidant, antiviral, and anticancer properties [66,67,68,69].

One of the most significant medicinal applications of spirulina is its role in supporting the immune system [23]. Studies have shown that the regular consumption of spirulina can boost both innate and adaptive immunity by enhancing the activity of NK cells, macrophages, and T cells [67]. This immunostimulatory effect makes spirulina valuable in preventing infections and improving immune responses in immunocompromised individuals [67].

Furthermore, spirulina has been reported to modulate the immune response during allergic reactions [101]. Research suggests that it may help reduce symptoms of allergic rhinitis by inhibiting the release of histamine from mast cells, offering a potential natural remedy for allergy sufferers [102]. These findings make spirulina a promising adjunct therapy for conditions that involve immune dysregulation, such as autoimmune diseases and allergies [101].

Spirulina and its derivatives have demonstrated antiviral effects, particularly against enveloped viruses such as HIV, HSV, and influenza [69]. Ca-SP, a sulphated polysaccharide isolated from spirulina, has been shown to inhibit viral replication by blocking the penetration of viruses into host cells [46]. 

Moreover, spirulina exhibits antimicrobial activity against pathogenic bacteria and fungi [103]. Its bioactive compounds, including phenolic acids and peptides, have been found to inhibit the growth of harmful microorganisms, making spirulina a potential candidate for developing antimicrobial therapies and natural preservatives in food and medicine [86].

The anticancer potential of spirulina has attracted considerable interest due to its ability to inhibit tumour growth and induce apoptosis in cancer cells [23]. Phycocyanin, a prominent compound in spirulina, has been shown to reduce oxidative stress and inflammation, two critical factors in cancer progression [104]. It also triggers cell death in various cancer cell lines by activating pro-apoptotic pathways and inhibiting angiogenesis, which is essential for tumour growth [66]. These findings suggest that spirulina could be explored as a complementary treatment for cancer, potentially reducing the side effects of conventional therapies like chemotherapy and radiation [80].

Given its lipid-lowering and anti-inflammatory properties, spirulina has shown promise in the management of chronic diseases, particularly cardiovascular diseases and type 2 diabetes [70,83]. Several studies have demonstrated that regular supplementation with spirulina can lead to reductions in total cholesterol, LDL cholesterol, and triglycerides, while simultaneously increasing HDL cholesterol levels, thereby reducing the risk of atherosclerosis and coronary heart disease [70]. Furthermore, its ability to improve glycaemic control makes spirulina a valuable supplement for individuals with diabetes, helping to stabilize their blood sugar levels and improve their insulin sensitivity [65].

In addition to its role in managing metabolic and cardiovascular diseases, spirulina has shown promise in addressing obesity by promoting weight loss and improving fat metabolism [105]. Its high protein content can help increase satiety and reduce overall calorie intake, making it a useful supplement in weight management programs [105].

Figure 1 summarizes the chemical composition, bioactive compounds, effects, and applications of spirulina microalga.

## 5. Future Perspectives and Research Directions

As the scientific community continues to explore the large potential of spirulina, its application in food, medicine, and animal feed is expected to expand [21,88]. Despite significant advancements, several areas remain underexplored, and future research will be crucial to unlocking new possibilities for this cyanobacterium in various industries [106].

One of the key areas for future development lies in optimizing the cultivation and production of spirulina to meet growing global demand [107]. Current methods of cultivation are often energy-intensive and require specific environmental conditions that may limit large-scale production [108]. Advances in biotechnological processes, such as photobioreactor design, automation, and genetic engineering, could significantly enhance spirulina yields while reducing costs and energy consumption [107]. Furthermore, optimizing cultivation conditions, such as nutrient availability, light exposure, and temperature, could increase the concentration of bioactive compounds like phycocyanin, polysaccharides, and omega-3 PUFAs [47,109]. Future research should focus on developing more sustainable and cost-effective cultivation strategies, including integrating Spirulina production with waste recycling systems and renewable energy sources [110].

While spirulina is already widely used as a dietary supplement and functional food ingredient, there is significant potential for expanding its applications in the feed sector [111]. Future research should explore the development of innovative food products that incorporate spirulina not only for its nutritional benefits but also for its therapeutic properties [23,48]. For instance, functional foods targeting specific health outcomes, such as cardiovascular health, diabetes management, or immune support, could be enhanced by incorporating spirulina in novel ways [66,71]. Research into biofortification and improving the bioavailability of spirulina’s nutrients through advanced food processing techniques could further expand its role in the global food supply [17,24].

The therapeutic potential of spirulina has been demonstrated in various in vitro and in vivo studies, particularly its antioxidant, anti-inflammatory, antiviral, and anticancer effects [73]. However, many of these findings have yet to be fully validated through clinical trials [112]. Future research should prioritize large-scale, randomized controlled trials (RCTs) to better understand the clinical efficacy and safety of spirulina in treating chronic diseases such as cancer, cardiovascular diseases, diabetes, and autoimmune disorders [112]. Furthermore, the mechanisms underlying spirulina’s bioactivities need to be elucidated at a molecular level, which could lead to the discovery of novel therapeutic compounds and drug development [23].

The interest in spirulina as a novel medical application is growing due to research on its bioactive compounds [23]. For example, spirulina-derived compounds could be investigated as potential drug candidates for antiviral therapies, especially in the context of emerging viral threats [66,69]. Its anti-inflammatory properties also hold promise for treating conditions such as arthritis, inflammatory bowel disease, and other chronic inflammatory disorders [68]. Additionally, with rising concerns about antibiotic resistance, spirulina’s antimicrobial properties could be further explored as natural alternatives to conventional antibiotics in medicine and veterinary care [58,69,113].

In the face of global challenges such as climate change, food insecurity, and malnutrition, spirulina offers a promising solution as a sustainable, nutrient-dense food source [7,41]. Future research should investigate its role in addressing these issues, particularly in developing regions where malnutrition and food shortages are prevalent [21]. As spirulina is relatively easy to cultivate and can grow in a variety of environments, it could be integrated into local agricultural systems to improve food security [114]. Additionally, exploring its potential as a resource for biofuel production, water purification, and carbon capture could further contribute to sustainability efforts [114].

Recent advances in genetic and metabolic engineering present exciting opportunities for enhancing spirulina’s bioactive properties [115]. By identifying key genes involved in the biosynthesis of valuable compounds such as phycocyanin, polysaccharides, and fatty acids, researchers can potentially enhance their production through targeted genetic modifications [116]. This could lead to the development of spirulina strains with higher yields of specific nutrients or bioactive compounds, making them more effective for both therapeutic and industrial purposes [116]. Moreover, genetic engineering could enable spirulina to be used as a bio-factory for producing pharmaceutical compounds, enzymes, and other high-value products [115].

In industrial biotechnology, spirulina has emerged as a versatile organism with a wide range of applications. Its high phycocyanin content, a vibrant blue pigment, not only makes it a valuable natural colorant in the food industry but also positions it as a sustainable alternative to synthetic dyes, particularly in confectionery, beverages, and cosmetics [23,117]. Beyond its use as a colorant, phycocyanin has been shown to possess potent antioxidant, anti-inflammatory, and immune-boosting properties, making spirulina a sought-after ingredient in nutraceuticals and functional foods.

Additionally, spirulina plays an important role in environmental biotechnology, particularly in the recovery of nutrients from wastewater generated by agriculture, aquaculture, and animal farming [118]. Its ability to absorb nitrogen and phosphorus from wastewater offers a dual benefit: it not only reduces environmental pollution but also creates a high-value biomass that can be repurposed as a biofertilizer or feed additive. This aligns with the growing interest in circular bioeconomy models, in which spirulina cultivation integrates waste recycling and resource optimization.

In biofuel production, spirulina is also gaining traction as a potential feedstock due to its rapid growth rate and ability to thrive in diverse environments [119]. Its lipid content can be harnessed to produce biodiesel, while the residual biomass can be converted into bioethanol or biogas, further contributing to sustainable energy solutions.

However, for both human and animal consumption, stringent safety standards must be maintained to ensure that spirulina is free from pathogens, heavy metals, and other contaminants. Achieving this requires advanced biotechnological processes such as controlled photobioreactor systems, optimized cultivation techniques, and effective post-harvest treatment methods. Despite the ongoing research in these areas, significant gaps remain in the literature, particularly regarding large-scale, cost-effective production methods that ensure consistent quality and safety. The potential for genetic engineering to enhance spirulina’s bioactive compound production and its ability to thrive in suboptimal conditions also represents an exciting frontier in industrial biotechnology. Future research should focus on optimizing cultivation conditions, improving bioavailability, and scaling up production processes to meet the growing global demand for spirulina across various sectors, including food, pharmaceuticals, cosmetics, and bioenergy.

Figure 2 summarizes this review, pointing out some of the strengths of, weaknesses of, opportunities for, and threats to spirulina utilization through an SWOT analysis. This figure highlights key aspects of spirulina’s potential applications in food, feed, and medicine, as well as the current challenges and future research opportunities for optimizing its use.

## 6. Conclusions

This review highlights the chemical composition and diverse bioactivities of spirulina microalga, positioning it as a highly valuable organism for applications in food, medicine, and animal feed. When compared to other microalgae such as *Chlorella vulgaris* and *Nannochloropsis oceanica*, spirulina stands out due to its higher protein content, greater abundance of essential vitamins, minerals, and fatty acids, and the presence of unique bioactive compounds like phycocyanin. These factors make spirulina an ideal candidate for nutritional supplementation and therapeutic use, particularly in areas where other microalgae have not been as extensively studied or have not shown the same level of versatility.

The antioxidant, anti-inflammatory, immunomodulatory, antiviral, anticancer, antidiabetic, and lipid-lowering properties of spirulina’s bioactive compounds, especially phycocyanin, polysaccharides, and carotenoids, have been well documented. These bioactivities offer significant potential for the prevention and treatment of chronic diseases. Moreover, spirulina’s role in animal feed is gaining increasing recognition, making it an essential component in sustainable animal nutrition, with demonstrated benefits in terms of growth performance, immune function, and disease resistance in livestock and aquaculture.

While substantial progress has been made in understanding spirulina’s bioactivities and applications, there is still a need for further research to optimize its cultivation methods, improve its bioavailability, and fully validate its therapeutic potential through clinical trials that focus on dose-effect relationships and safety. Future research should also explore novel applications in biotechnology, environmental sustainability, and medicine, as spirulina holds great promise as a key contributor to addressing global challenges in health, food security, and animal feed sustainability.

## Figures and Tables

**Figure 1 foods-13-03656-f001:**
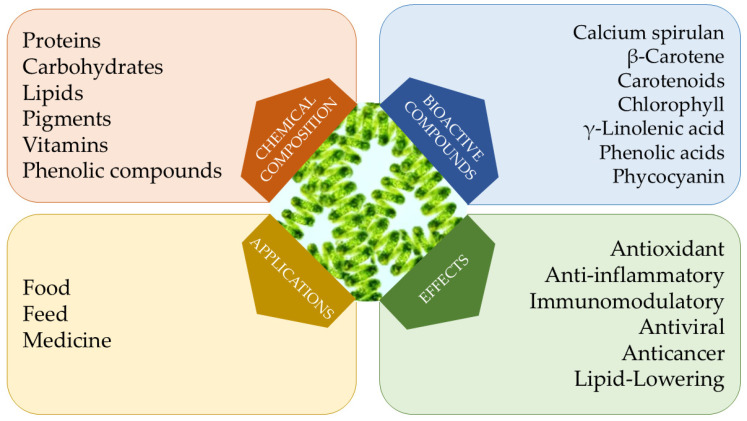
Chemical composition, bioactive compounds, effects, and applications of spirulina microalga.

**Figure 2 foods-13-03656-f002:**
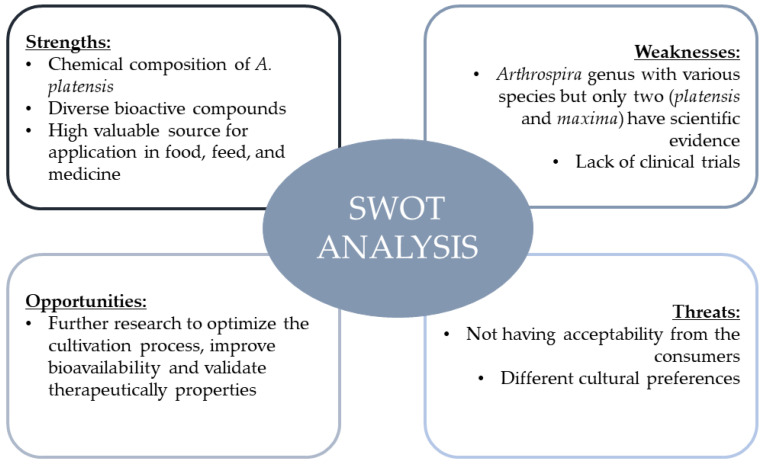
SWOT analysis summarizing the strengths, weaknesses, opportunities, and threats associated with *Spirulina* utilization.

**Table 1 foods-13-03656-t001:** Spirulina’s major chemical compounds and associated bioactivities, with entries ordered by the alphabetical name of the compound.

Pre-Treatment	Operation Conditions	Advantages	Disadvantages	Main Reference(s)
Bead milling	Algal suspension with 0.5 mm zirconium beads (1 bead per 1 mL) for 30 min at 2000 rpm in a shaking orbit	Effective in alga cell disintegration and easy method to perform Used in aquaculture to improve digestibility	Efficacy depends on the load and diameter of the beads	[25,26,31,32,33]
Extrusion	Microalgae at 34 bars and 114 °C for 3–7 s, followed by water mL/min and drying at 120 °C for 8–10 min	Successfully breaks microalga cell wall structure Applied on a large scale	High pressure and temperature may cause denaturation and aggregation of proteins and lead to irreversible modifications	[25,26,34]
Freeze-drying	Algal biomass lyophilisation for 24 h, after overnight freezing at −80 °C	Easy method for lipid extraction Does not affect cellular components	Methodology is expensive	[25,26,35,36]
Heating	Exposing dried microalgae to heating at 70 °C for 30 min on a stove	Improves proteins’ microalgae accessibility as it modifies proteins in peptides and amino acids	High temperatures may lead to the formation of complexes that hinder solvent extraction	[25,26,31,37,38]
Microwave	The keep warm setting is used until microalga suspension boils	Combination of electromagnetic field and heat Applied on a large scale	Low extraction yield	[25,26,31,33]
Sonication	Ultrasound device regulated for seven cycles at 70% potency, 200 W, and 20 kHz for 15 min	Waves cause cell wall disruption by cavitation within the cell wall	The cavitation process can destroy organic matter and produce shear force, producing reactive radicals	[25,26,32,39]

**Table 2 foods-13-03656-t002:** Spirulina’s major chemical compounds and associated bioactivities, with entries ordered by the alphabetical name of the compound.

Compound Name or Type	Type of Bioactivity(ies) (Underlying Mechanism)	Main Reference(s)
Calcium spirulan (Ca-SP; see also sulphated polysaccharides)	- Antiviral (inhibits viral replication and entry into host cells)	[46,85]
β-Carotene	- Antioxidant (protects against oxidative damage) - Anticancer (prevents DNA damage)	[51,81,85]
Carotenoids	- Antioxidant (neutralizes singlet oxygen and reactive oxygen species) - Anticancer (reduces risk of carcinogenesis)	[23,51]
Chlorophyll	- Antioxidant (supports detoxification and protects tissues)	[23,72]
γ-Linolenic acid (18:3n − 6)	- Lipid-lowering (regulates lipid metabolism) - Antidiabetic (improves insulin sensitivity, reduces blood glucose levels)	[65,70,83,85]
Phenolic acids	- Antibiotic activity (against pathogenic bacteria and fungi)	[86]
Polyphenols	- Antioxidant - Anti-inflammatory	[55]
Phycocyanin	- Antioxidant - Anti-inflammatory - Immunomodulatory - Anticancer - Antiviral (reduces viral replication) - Reduces pro-inflammatory cytokines (inhibits COX-2) - Induces apoptosis in cancer cells (inhibits angiogenesis) - Lipid-lowering (regulates lipid metabolism) - Antidiabetic (improves insulin sensitivity and reduces blood glucose levels)	[44,65,69,70,73,80,85]
Polysaccharides	- Anti-inflammatory (suppresses NF-κB activation) - Immunomodulatory (enhances cytokine production) - Antidiabetic (improves insulin resistance and reduces blood glucose levels)	[23,65,74]
Sulphated polysaccharides (see also Ca-SP)	- Antiviral (inhibits enveloped virus replication) - Anticancer (anti-proliferation of cancer cells)	[46,82,85]

## Data Availability

No new data were created or analyzed in this study. Data sharing is not applicable to this article.

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
