# Peer review of "Chemical Composition, Bioactivities, and Applications of Spirulina (Limnospira platensis) in Food, Feed, and Medicine"

_foods, 2024, doi:10.3390/foods13223656_

Round 1

Reviewer 1 Report

Comments and Suggestions for Authors

1.Bibliometrics is suggested in this review rather than simple statement;

2.There must be some figures applied for summarizing those key compounds, findings, products and activity mechanisms with copyright.

3.Other objects in Limnospira genus with similar constituents and bioactivities should also be introduced in Introduction with moderate text length. 

4.More details can be mentioned in the tables, e.g., operation conditions of pre-treatment in Table 1.

5.SWOT analysis is suggested after the main contents.

Author Response

The authors express their gratitude to the Editor and Reviewers for their throughout review and constructive comments on our manuscript. We have addressed all concerns and incorporated the suggested changes as described below.

1.Bibliometrics is suggested in this review rather than simple statement.

Reply: Thank you for your valuable suggestion regarding the use of bibliometrics. We have taken this into account and enriched our review with bibliometric analysis to provide a more quantitative perspective on the research related to Spirulina (Limnospira platensis).

2.There must be some figures applied for summarizing those key compounds, findings, products and activity mechanisms with copyright.

Reply: Thank you for your suggestion regarding the use of figures to summarize key aspects of our review. In the revised manuscript, we have added Figure 1, which provides a comprehensive overview of the chemical composition, bioactive compounds, effects, and applications of Spirulina microalga. This visual summary helps to clarify key findings, activity mechanisms, and potential applications. Additionally, we have included Figure 2, which presents a SWOT analysis of the research topic, offering further insight into the strengths, weaknesses, opportunities, and threats associated with Spirulina research.

3.Other objects in Limnospira genus with similar constituents and bioactivities should also be introduced in Introduction with moderate text length.

Reply: Thank you for your suggestion. In the revised manuscript, we have added a paragraph to the Introduction that discusses other species within the Limnospira genus, including L. maxima, L. fusiformis, and L. laxissima. This addition provides context on similar constituents and bioactivities, thereby enriching the background information with a broader perspective on related species.

4.More details can be mentioned in the tables, e.g., operation conditions of pre-treatment in Table 1.

Reply: Thank you for your suggestion. In the revised manuscript, we have expanded Table 1 to include detailed operation conditions for each pre-treatment method. These additions provide more comprehensive information, enhancing the clarity and utility of the table for readers.

5.SWOT analysis is suggested after the main contents.

Reply: Thank you for your suggestion. In the revised manuscript, we have added Figure 2, which presents a SWOT analysis summarizing the strengths, weaknesses, opportunities and threats related to the topic. This figure is included after the main content to provide a strategic overview and further enrich the discussion.

Reviewer 2 Report

Comments and Suggestions for Authors

The presented manuscript: "Chemical Composition, Bioactivities and Applications of Spirulina (Limnospira platensis) in Food, Feed, and Medicine" represents a solid review for the biochemistry and bioactivity of Spirulina that makes this species very popular in the food and pharmaceutical industry.

The following changes are recommended and some clarifications should be made:

There are too many keywords. They can be reduced.

Line 153: The part for secondary metabolites is shortly presented and only phenolics as a major group and two phenolic acids are mentioned. The authors should paid more attention to this part and to include more phenolics or other secondary metabolites identified in Spirulina since the bioactivities are generally related to this type of secondary metabolites.

Line 167: Are there studies reporting that phenolics contributed to the antioxidant activity of Spirulina extracts?

Line 205: Please avoid starting a sentence with an abbreviation. This should be implemented throughout the manuscript.

Author Response

The authors express their gratitude to the Editor and Reviewers for their throughout review and constructive comments on our manuscript. We have addressed all concerns and incorporated the suggested changes as described below.

The presented manuscript: "Chemical Composition, Bioactivities and Applications of Spirulina (Limnospira platensis) in Food, Feed, and Medicine" represents a solid review for the biochemistry and bioactivity of Spirulina that makes this species very popular in the food and pharmaceutical industry.

Reply: Thank you for your positive feedback on our manuscript. We appreciate your comments and have carefully addressed all the suggestions provided to further enhance the quality and clarity of the review.

The following changes are recommended and some clarifications should be made:

There are too many keywords. They can be reduced.

Reply: Thank you for your suggestion. In the reviewed manuscript, we decreased the number of keywords to 8.

Line 153: The part for secondary metabolites is shortly presented and only phenolics as a major group and two phenolic acids are mentioned. The authors should paid more attention to this part and to include more phenolics or other secondary metabolites identified in Spirulina since the bioactivities are generally related to this type of secondary metabolites.

Reply: Thank you for your suggestion. In the revised manuscript, we have expanded Section 2.6 on secondary metabolites to provide a more detailed discussion of phenolic compounds, including additional examples and their associated bioactivities. We have also elaborated on their health-beneficial contributions to enhance the understanding of their importance. These revisions can be found on page 5, lines 175-191.

Line 167: Are there studies reporting that phenolics contributed to the antioxidant activity of Spirulina extracts?

Reply: Thank you for your comment. In the reviewed manuscript, we added information about studies that report the effects of phenolic compounds to the antioxidant activity of Spirulina extracts.

Line 205: Please avoid starting a sentence with an abbreviation. This should be implemented throughout the manuscript.

Reply: Thank you for your suggestion. We have revised the manuscript to ensure that no sentences begin with abbreviations, implementing this change consistently throughout the text for improved readability and clarity.

Round 2

Reviewer 1 Report

Comments and Suggestions for Authors

Accept in present form